# The Impact of Groundwater Model Parametrization on Calibration Fit and Prediction Accuracy—Assessment in the Form of a Post-Audit at the SLOVNAFT Oil Refinery Site, in Slovakia

Martin Zatlakovič [1,*], Dávid Krčmář [1], Kamila Hodasová [1], Ondra Sracek [2], Štefan Marenčák [3], Ľubica Durdiaková [3] and Alexander Bugár [1,3]

1   Department of Engineering Geology, Hydrogeology and Applied Geophysics, Faculty of Natural Science, Comenius University in Bratislava, Ilkovičova 3278/6, 84215 Bratislava, Slovakia
2   Department of Geology, Faculty of Science, Palacký University Olomouc, 17. listopadu 1192/12, 771 46 Olomouc, Czech Republic
3   VÚRUP, a.s., Vlčie Hrdlo, P.O. Box 50, 82003 Bratislava, Slovakia
*   Correspondence: martin.zatlakovic@uniba.sk

**Abstract:** The present work is focused on the effect of increasing model complexity on calibration fit and prediction accuracy. Groundwater flow was numerically simulated at a field site with a hydraulic groundwater protection system in operation with many pumping and observation wells at the site of the Slovnaft refinery in southwestern Slovakia. The adjusted parameters during the calibration included hydraulic conductivity, as well as recharge, evapotranspiration, and riverbed conductance. Four model scenarios were built (V1–V4) within the model calibration for the conditions in the year 2008, with increasing complexity mainly within artificial K-field zonation, which was created and step-wise upgraded based on groundwater head residuals' distribution. Selected descriptive statistics were evaluated together with chosen information criteria after the models were calibrated. Subsequently, the real predictive accuracy of individual calibrated scenarios was evaluated for conditions in the year 2019 in the form of a post-audit. Within the overall evaluation, the calibration fit increased with increased parameterization complexity. However, the Akaike information criterion, corrected Akaike information criterion, and Bayesian information criterion detected opposite trends for model predictability. A post-audit of prediction accuracy revealed a significant improvement of the V2, V3, and V4 scenarios against the simplest V1 scenario. However, among the V2–V4 scenarios, the degree of prediction accuracy improvement was almost insignificant. The level of effort spent on V3 and V4 parameterization seems disproportionate to the benefit of a negligible improvement in prediction accuracy. Groundwater flow path analysis showed that similarly successful scenarios (measured by prediction accuracy) can generate very different groundwater pathlines.

**Keywords:** numerical modeling; model predictability; calibration; prediction; information criteria

## 1. Introduction

In groundwater modeling, great attention is paid to the calibration and the predictive accuracy of models in relation to their complexity, especially to the distribution of hydraulic conductivity K. There are three key parameters of the K-field calibration: the shape and distribution of K zones, thickness of the aquifer within a particular zone, and optimized K values in each zone. The most important issue is the determination of spatial K distribution since the lithological data are generally insufficient due to their point character. Typically, we sample less than one millionth of the material we are characterizing [1]. The K values are also affected by the thickness of an aquifer generally interpolated from only limited point data. The attempt of K-field calibration with artificial zonation based on residuals of

water level distribution has only limited validity, but may result in a better approximation of a real system in terms of calibration and predictions.

When simulating most natural systems, commonly there are alternative plausible models. For example, alternative models of a groundwater system may be developed due to uncertainty associated with the following: (1) The structure and character of boundary conditions. (2) Relevant processes. (3) The spatial and temporal distribution of system characteristics such as hydraulic conductivity, recharge, reaction coefficients, and so on, including alternatives based on different ideas about the deposition and deformation of geologic materials. (4) The inclusion or exclusion of transients associated with, for example, pumping rates, source concentrations, recharge, and so on [2]. A framework for dealing with uncertainty due to model structure error was introduced, e.g., by [3].

To determine the most probable modeling scenarios, selected descriptive statistics and the Akaike information criterion (AIC) [4], Bayesian information criterion (BIC) [5], and corrected Akaike information criterion (AICc) [6] can be used in the post-calibration stage. Those and other criteria based on information theory have been widely implemented in the groundwater modeling field. An example can be cited in [7–15], beside many others. However, despite their broad use in modeling, the foundations of the AIC, AICc, and BIC, which penalize the likelihoods in order to select the simplest model, are, in general, according to [7], poorly understood. There have also been various averaging schemes developed to deal with multiple models (variants) in order to generate the most probable scenario [2,8,9]. Weighting of models within such schemes can be based on mentioned information criteria. After the probability of models is set, the real prediction accuracy can be evaluated for chosen scenarios in the form of a post-audit [16,17], in accordance with the statement of [18], i.e., that in any event, the accuracy of the prediction cannot be assessed until the predicted period of time has passed.

Ref. [4] saw that the difficulty of constructing an adequate model based on the information provided by a finite number of observations was not fully recognized (by professionals). Therefore, he introduced the AIC, which provides a mathematical formulation of the principle of parsimony in the field of model construction [4]. Ref. [5] introduced the BIC, which qualitatively, like the AIC, gives a mathematical formulation of the principle of parsimony in model building. Quantitatively, the BIC procedure leans more towards lower-dimensional models. For a large number of observations, the procedures differ markedly from each other [5]. AICc, which is another information criterion that was introduced by [6], is a bias-corrected version of AIC for nonlinear regression and autoregressive time series models. In view of the theoretical and simulation results, the AICc should be used routinely instead of the AIC for regression and autoregressive model selection [6].

Since the end of the last century, one can find the prevailing opinion for inverse modeling methodology, in which it was emphasized, e.g., by [19,20], to begin calibration estimation with very few parameters that together represent most of the features of interest, and to increase the complexity of the parameterization slowly. The importance of keeping a model simple, the principle of parsimony [2,8–11,19,20], is demonstrated by noting that more complex models generally fit the observations more closely, yet they can have greater prediction error compared to simpler models [20] and others.

It is not easy to decide whether to use a more complex model or simple calculations combined with expert judgement. Ref. [21] introduced several examples of opinions regarding model simplicity vs. complexity from experts in the field. Some authors advocate complexity, e.g., [22], while others are not convinced about its definite benefits, e.g., [20,23,24]. Ref. [25] preferred constructing models that were a compromise between the effort of the professional's perfection expressed in very complex models and oversimplified models which strive for speed and efficiency. He also saw the real role of models in extracting the maximum amount of information from the data and minimizing the uncertainty, both via history matching approaches. Ref. [26] provided a theoretical analysis of the model simplification process, yielding insights into the costs of model simplification, as well as into how some of these costs may be reduced. They claim that modern envi-

ronmental management and decision making is based on the use of increasingly complex numerical models. They see the advantage of complex models in the possibility of the expert knowledge application within them. The disadvantage of such models lies in the problematic calibration and analysis of their prediction uncertainty. On the other hand, many system and process details on which uncertainty may depend are, by design, omitted from simple models. According to the authors, this can lead to underestimation of the uncertainty associated with many predictions of management interest.

Our study site is located in southwest Slovakia close to Bratislava (Figure 1), in the proximity of the Danube River. In the past, the site was strongly affected by contamination from petroleum products from the Slovnaft refinery and the hydraulic protection system connected to the monitoring system has been operating there for several decades. This means that long-term monitoring data are available for groundwater modeling.

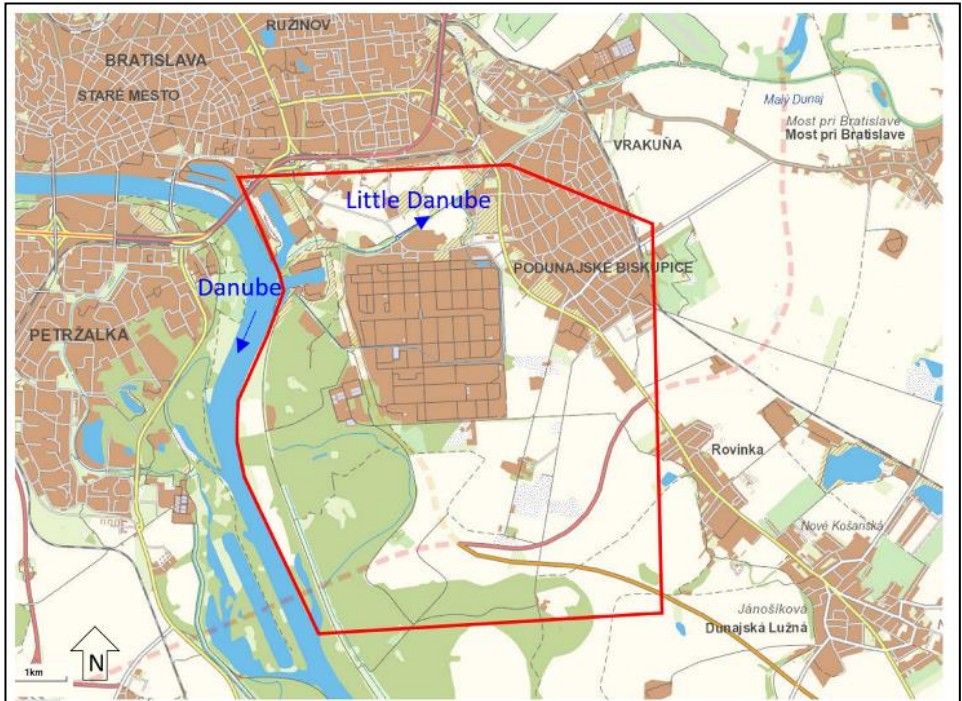

**Figure 1.** Location of the model domain in Bratislava, Slovakia, near the Danube River.

The main objective of this work is to illustrate the impact of simple to medium model complexity (parametrization) on the simulated groundwater head agreement with field observations within calibration and prediction processes. Additionally, this objective also includes its impact on calculated groundwater pathlines at the study site.

## 2. Materials and Methods

The study site is situated in Bratislava, in the southwest of Slovakia, close to the Danube River (Figure 1). The model area is about 39 km$^2$ and is situated on the northernmost part of a river island bordered by the Danube's main course (SW border of the island) and its branch—the Little Danube River (NE border of the island). There are fluvial sand and gravel deposits form the main and most significant part of the investigated aquifer. The thickness of these deposits ranges between 12 m on the NW and 45 m on the east part of the modeled area. Under the sand and gravel Quaternary layers, Neogene fine sand occurs with significantly lower hydraulic conductivity. The base of these permeable units consists of Neogene clays (Figures 2 and 3). In the central part of the model domain, the SLOVNAFT refinery area is situated with the pumping wells of its groundwater hydraulic protection system (GWHP) operated by the VÚRUP company. The GWHP system prevents the spreading of polluted groundwater outside the refinery area [27] and represents an

important model stressor. The number of pumping wells vary depending on the actual conditions but is usually around 70 (Figure 4). Additionally, the actual pumping rates of individual wells vary depending on actual conditions and are measured daily. The refinery area and its surroundings are highly populated by observation wells (875 at the end of 2019). The monitored parameters consist of a hydraulic head as well as the groundwater quality. The head observation frequency depends on the position of a particular observation well. Inside the refinery area, monitoring is conducted either once a day or once a week, depending on the occurrence of GW pollution in a given well. In the vicinity of the refinery area, it is four times a year, while in the more distant surroundings of the refinery, it is twice a year.

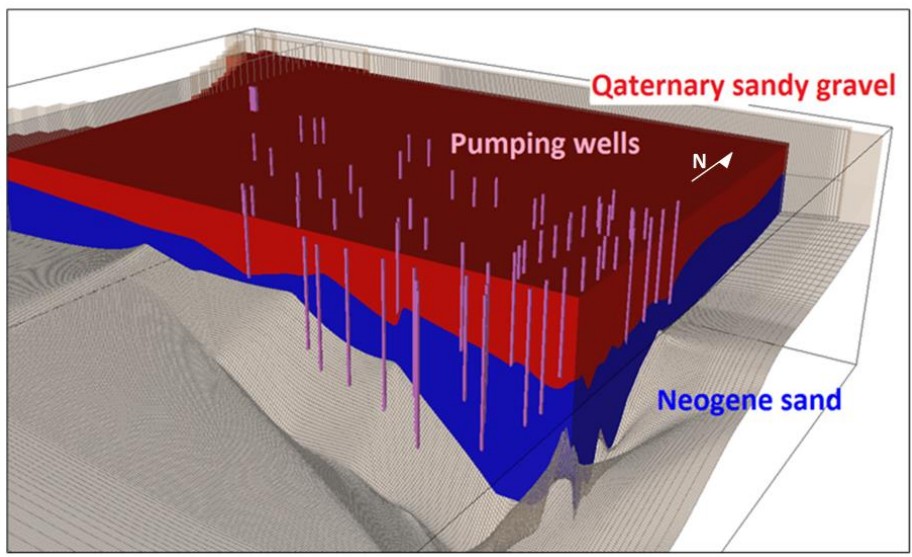

**Figure 2.** Block diagram of geological settings with the principal aquifer layers and pumping wells depiction (vertical exaggeration 1:20).

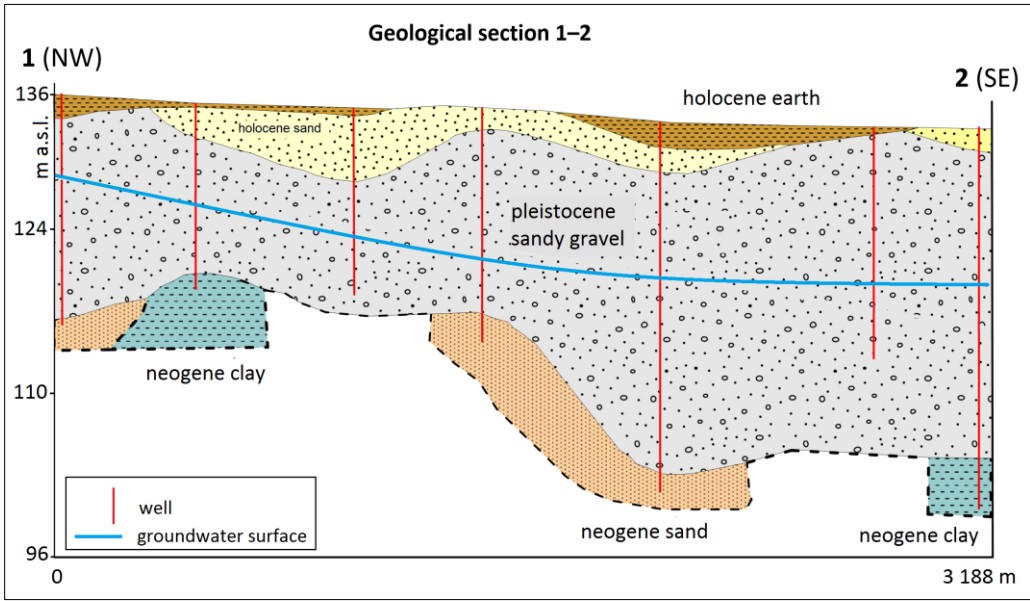

**Figure 3.** Geological section of modeled area central part (the refinery area) with approximate groundwater surface after [27], the section line depicted in Figure 4.

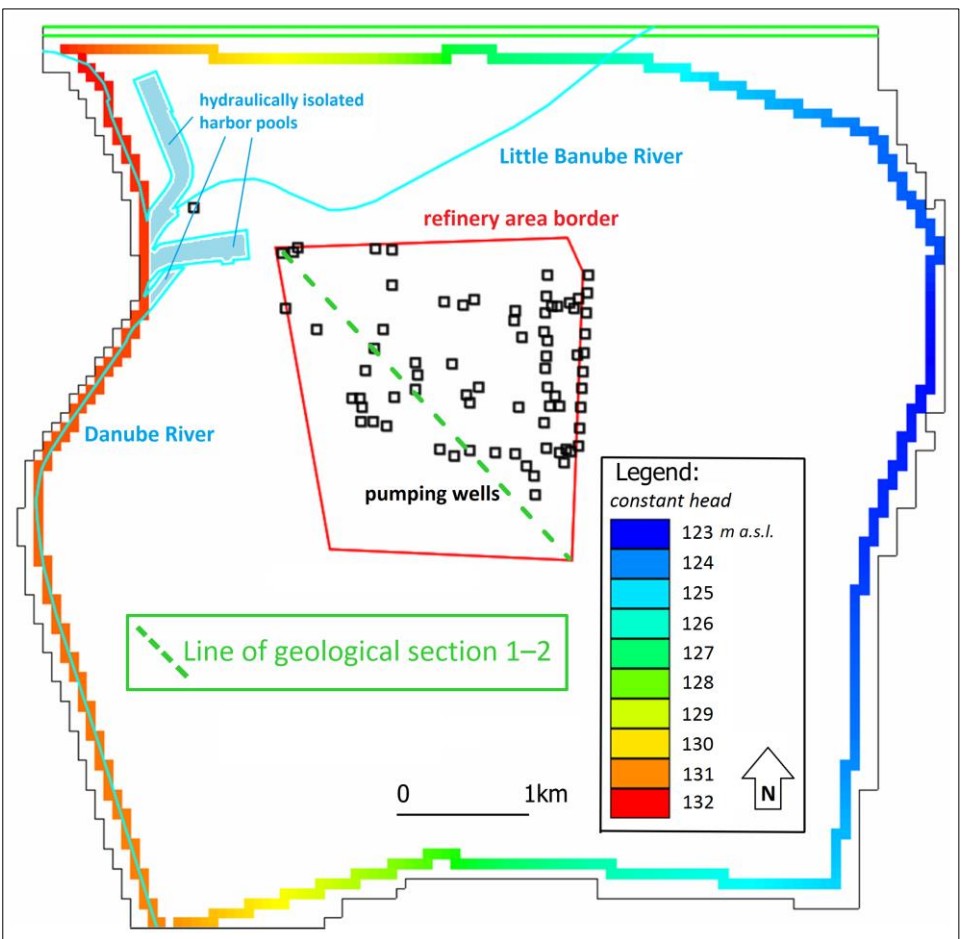

**Figure 4.** Constant head boundary level, pumping wells, surface water bodies and situation of the geological section 1–2 in the model domain.

### 2.1. Model Settings

Model geological settings are based on the lithological data from the GWHP well logs. The geological conditions were simplified into two aquifers, Quaternary and Neogene, with underlying Neogene clays representing an aquiclude. The strata interfaces are spatially interpolated by the natural neighbor method. The hydraulic conductivity within these aquifers is assumed to be up to $5 \times 10^{-2}$ m·s$^{-1}$, and horizontal hydraulic conductivity is regarded as isotropic (Kx = Ky). The vertical hydraulic conductivity is computed by the formula Kz = Kxy/5. Groundwater within the modeled area is under unconfined conditions. Model hydrological settings are represented by the long-term (1961–1990) annual precipitation, which averages around 600 mm [28], as well as the long-term (1961–1990) annual actual evapotranspiration, which averages around 450 mm [28]. Therefore, around 150 mm of precipitation is left annually for infiltration on average. Recharge of the modeled aquifer is predominantly secured by water infiltration from the Danube River. This river also defines the western border of the model domain, and with its average discharge of around 2000 m$^3$·s$^{-1}$, it represents one of the model's constant head boundary conditions. Its branch, the Little Danube River, is represented as a 3rd type boundary condition and is situated on the northern part of the model domain (Figure 4). Groundwater within the study area is intensively extracted by the pumping wells of the GWHP system and is discharged outside of the model domain after treatment. The overall pumping rate varies based on the GWHP system dynamic operation. Throughout the years, new pumping wells have also been introduced, since some of the old ones have been deactivated.

Boundary condition of the 1st type H = f(x,y,z,t). On the western border of the model area, the Danube is represented by a 1st type boundary condition (Figure 4). The height

of the water level is obtained by a linear interpolation between two gauging stations that are located outside but near the model domain. The interpolation was performed between averaged water levels for the relevant period. The Danube water is directly connected with the groundwater. With regards to its size, the river represents a conceptually infinite source of infiltrated water. The other boundaries of the modeled area (northern, eastern, southern) are also represented by 1st type boundary conditions (Figure 4). In this case, the constant hydraulic head is computed by linear interpolation of the averaged groundwater head observed in the monitored wells situated along the model boundary (15 wells, which are regularly spaced along the model area boundaries). This boundary condition is simulated by the Time-Variant Specified-Head package (CHD) of MODFLOW-2005 [29].

Boundary condition of the 2nd type Q = f(x,y,z,t)—the recharge defined by the difference between precipitation and evapotranspiration is variously represented within individual model scenarios as described further below (Section 2.3). Groundwater extraction by the pumping wells of the GWHP system is represented by averaged daily measured pumping rates at individual wells for the given time period. This boundary condition is simulated by the well (WEL), recharge (RCH), and EVT module of MODFLOW-2005 (Harbaugh 2005).

Boundary condition of the 3rd type Q = f(H)—leakage between groundwater and surface water of the Little Danube River is simulated by the river module (RIV) [29]. The infiltration rate is driven by the difference between groundwater and surface water levels, as well as by the conductance of the riverbed, which was calibrated as a parameter in given calibration scenarios.

Four alternative steady-state groundwater model scenarios (V1, V2, V3, V4) were developed which differ in the representation of hydraulic conductivity, recharge, evapotranspiration, and riverbed conductance. All the groundwater flow models utilize the numerical finite-difference method. The following MODFLOW-2005 software modules were used: layer property flow (LPF), preconditioned conjugate gradient (PCG), CHD, RCH, EVT, WEL, RIV, and hydraulic head observations (HOB). The first layer within the LPF module is defined as convertible (unconfined with the possibility to be automatically converted to confined when fully saturated), while other layers are defined as confined because of their fully saturated behavior in all simulations. This procedure is standard in the LPF package within the MODFLOW-2005 program [29]. Intercell hydraulic conductivity is calculated by the harmonic mean method.

*2.2. Model Calibration and Prediction*

Within the inverse modeling by the UCODE software [30], the hydraulic conductivity of the first Quaternary layer was found to be the most influential parameter. The spatial distribution of K values was optimized for the predefined K zones. Each K zone acts as an individual calibration parameter. The distribution of the K zones was defined manually within all scenarios, starting with homogeneous K for individual layers in model V1. Afterwards, the areas of Quaternary sediments with groupings of positive or negative head residuals, i.e., the areas of insufficient K distribution, which resulted in a constant tendency to compute too high or too low groundwater heads by the model, were split into smaller zones. The number of K zones gradually increased within individual calibration model variants (V2–V4). All underlying Neogene layers were left as one adjusted parameter. The conductance of the Little Danube riverbed was calibrated in the V3 and V4 models. Regional water recharge by precipitation and actual evapotranspiration were optimized using various approaches within individual model scenarios. A modified Gauss–Newton method, also called the Marquardt–Levenberg method of nonlinear regression [19], was used for the parameter optimization. The weights of the individual hydraulic head observations, which was the only observed parameter, had a uniform value of "1" due to the assumption of its equivalent error variation range. The set objectives of the presented work were accomplished by model calibration of each used scenario for conditions of the entire year of 2008 (averaged observation values used), followed by the comparison of

real prediction accuracy of individual scenarios in the conditions of the entire year of 2019 (averaged observation value used). For model fit evaluation, the calculated GW head (SIM) and observed GW head (OBS) were used in various statistical and graphical procedures. The observed heads also served as reference values in the calibration process. Residuals (RES)—the differences between OBS and SIM—provided a basic quantitative expression of a given match. The sum of squared residuals or objective function (OF) is the overall expression of the SIM and OBS agreement, as well as the residuals' root mean square error (RMSE) value and the averaged absolute residuals (ABS AVG RES). The theoretical relative prediction accuracy of the models was evaluated by the information criteria AIC, AICc, and BIC and compared with the real models' performance of predictions for 2019 conditions. The real prediction accuracy of the models was evaluated by the post-audits in the same manner as for the calibration. The information criteria were calculated as follows (Equation (1) from [4] in [19], Equation (2) from [6], Equation (3) from [5] in [19]):

$$AIC = OF + 2 \, nPAR \tag{1}$$

$$AICc = AIC + 2 \, (nPAR + 1) \, (nPAR + 2)/(HOB - nPAR - 2) \tag{2}$$

$$BIC = OF + nPAR \, \ln(HOB) \tag{3}$$

OF: objective function (sum of squared residuals),
HOB: groundwater head observations number,
nPAR: number of adjusted parameters.

Since the value of the given information criteria is relative, in equations 1, 2, and 3 the maximum likelihood function was substituted by the objective function and the constant terms were dropped from their original formulations in accordance with [12]. For all of the information criteria, smaller relative values indicate a more accurate model. The OF term in the equation of both information criteria is the measure of overall fit. In general, its value decreases with an increase in the number of the parameters. The term nPAR can be seen as a "penalty" for "too many" parameters. It is evident that the criteria favor a compromise between a good model fit and a small number of parameters. Model scenarios V1–V4 were calibrated to the dataset of hydraulic heads computed as the averaged observations for the entire year of 2008. The prediction accuracy of each calibrated model scenario was evaluated by assessment of their performance in 2019 conditions, in the form of a post-audit. The differences between the values of the most fundamental physical conditions in the 2008 calibration period and the 2019 prediction period introduced in Table 1 and Figure 5 are assumed to be sufficient to consider the performance of calibrated (for 2008) models in 2019 as a prediction in the true sense, i.e., the performance in extrapolated conditions. Finally, simple GW flow analysis using MODPATH 6 software [31] was performed to compare pathlines of virtual particles from virtual sources of individual modeled scenarios. This analysis was applied at the most complex part of the modeled domain—at the refinery area with the presence of pumping wells.

**Table 1.** Comparison of the values of main features influencing modeled conditions between the years 2008 and 2019.

| Feature | Units | 2008 | 2019 | ABS Delta |
|---------|-------|------|------|-----------|
| AVG OBS | m a.s.l. | 124.27 | 123.69 | 0.58 |
| AVG RIV 1 | m a.s.l. | 131.83 | 131.99 | 0.16 |
| AVG RIV 2 | m a.s.l. | 130.91 | 130.87 | 0.04 |
| AVG Q $_{pumping}$ | m$^3 \cdot$s$^{-1}$ | 0.916 | 1.009 | 0.093 |

Notes: AVG OBS: average of observed groundwater head; AVG RIV: averaged Danube River level in used gaging stations; AVG Q pump: averaged overall pumping rate at modeled site; ABS delta: absolute value of the difference between 2008 and 2019 period.

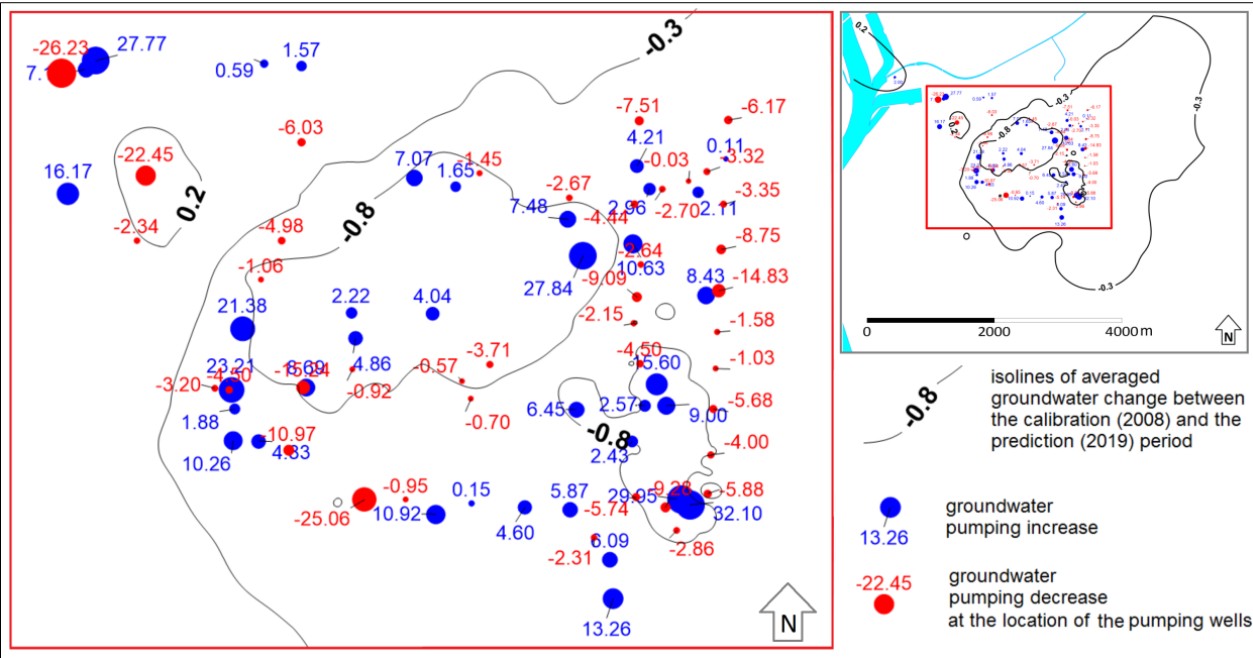

**Figure 5.** Differences in averaged observed groundwater head (contours) and averaged pumping rates (proportional dots), respectively, between the period of prediction (2019) and the calibration period (2008) (2019 minus 2008). The whole model domain view (**left**) and detail view on the central model area (**right**).

### 2.3. Conceptual Approach in Individual Model Scenarios

The first (V1) variant (Figure 6a) was calibrated with homogeneous K within the individual layers of the model. The calibrated parameters were the homogeneous K of the 1st aquifer and the homogeneous K for the 2nd aquifer. Actual evapotranspiration and regional recharge by precipitation were homogeneous and not calibrated, with the applied values taken from the Atlas of Landscape of the Slovak Republic [28]. The average annual EVT for a given locality reaches 450 mm, while the annual average RCH for a given locality reaches 550–600 mm. A value of 600 mm was used. In the V1 variant, the Little Danube River was assumed without any connection with groundwater.

In the second (V2) variant (Figure 6b), a software-automated zonal K calibration was performed within the first layer, formed of Quaternary highly permeable sand and gravels (39 K zones as parameters). The K of the second layer of Neogene sand was optimized as one parameter. The rate of water supply by recharge to the system was optimized through zonal calibration of evapotranspiration (Figure 7). EVT was adjusted within three zones—forest areas, urban areas, and agriculture field areas. The Little Danube River was represented in the same manner as in the V1 variant.

The third (V3) variant (Figure 6c) represented the reduction of the computational grid density to approximately one-quarter of the active cells number compared to all other variants. The K calibration zones have a different shape and are more refined. The total number of K parameters (zones) is 135. The conductance of the Little Danube riverbed sediments was automatically calibrated (Figure 8a). The regional recharge by precipitation was automatically calibrated using one parameter value applied for the whole model domain, which was conceptually taken as the difference between RCH and EVT. Thus, EVT was not calibrated as a distinct parameter. The initially entered value of annual average precipitation was reduced by the value of the average annual actual EVT based on data from the Atlas of Landscape of the Slovak Republic [28]. The calculated difference (150 mm·y$^{-1}$) is the value of $4.76 \times 10^{-9}$ m·s$^{-1}$.

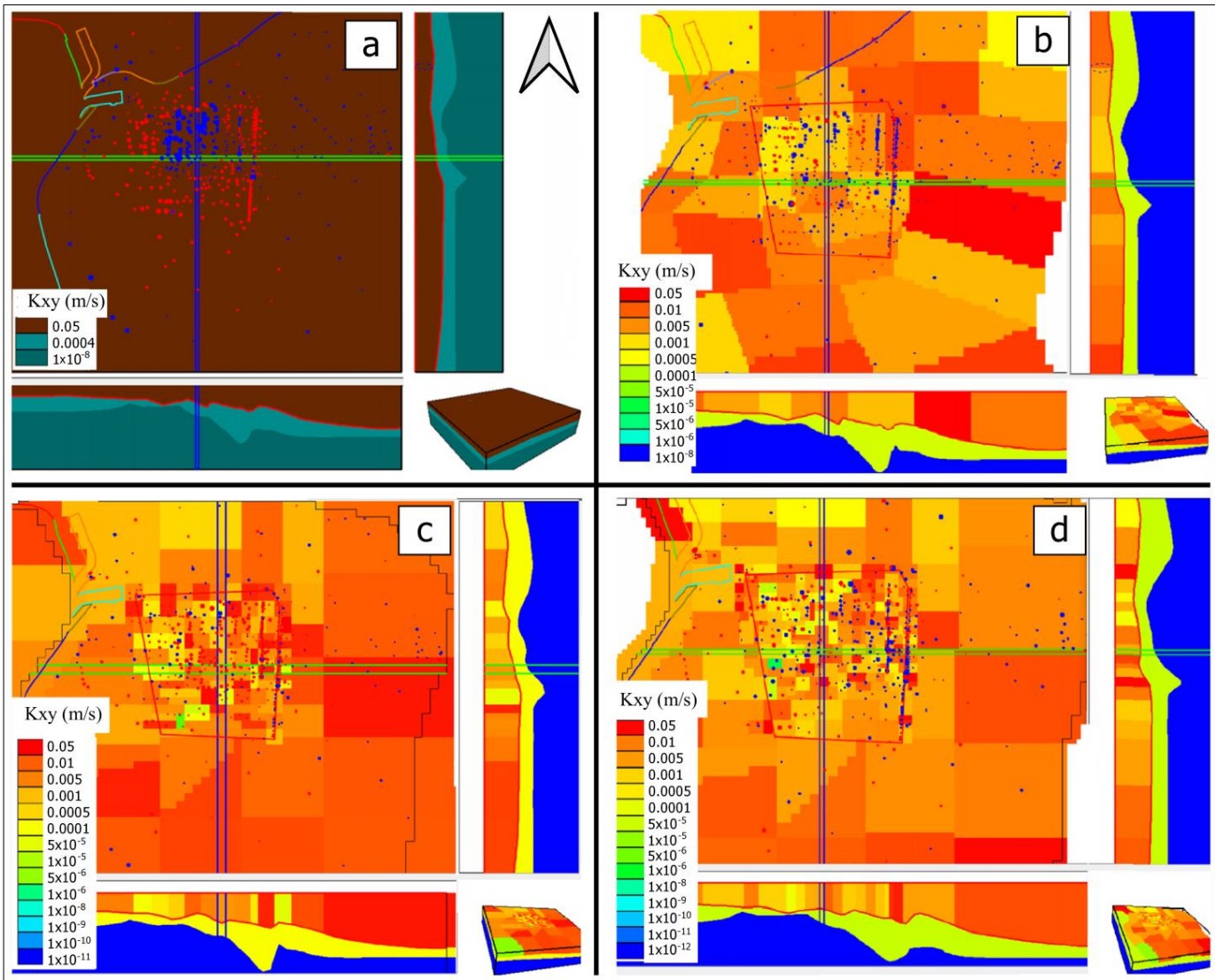

**Figure 6.** Horizontal hydraulic conductivity (Kxy) zonation within individual model variants ((**a**)—V1, (**b**)—V2, (**c**)—V3, (**d**)—V4) with the optimized Kxy values after calibration (planar view and vertical cross sections along the blue and green double lines).

The fourth (V4) variant (Figure 6d) differed from V3 by the higher density of the calculation grid and higher parameterization of K spatial distribution (251 K parameters). The riverbed sediments conductance and regional recharge parametrization approaches were similar to the V3 variant (Figure 8b).

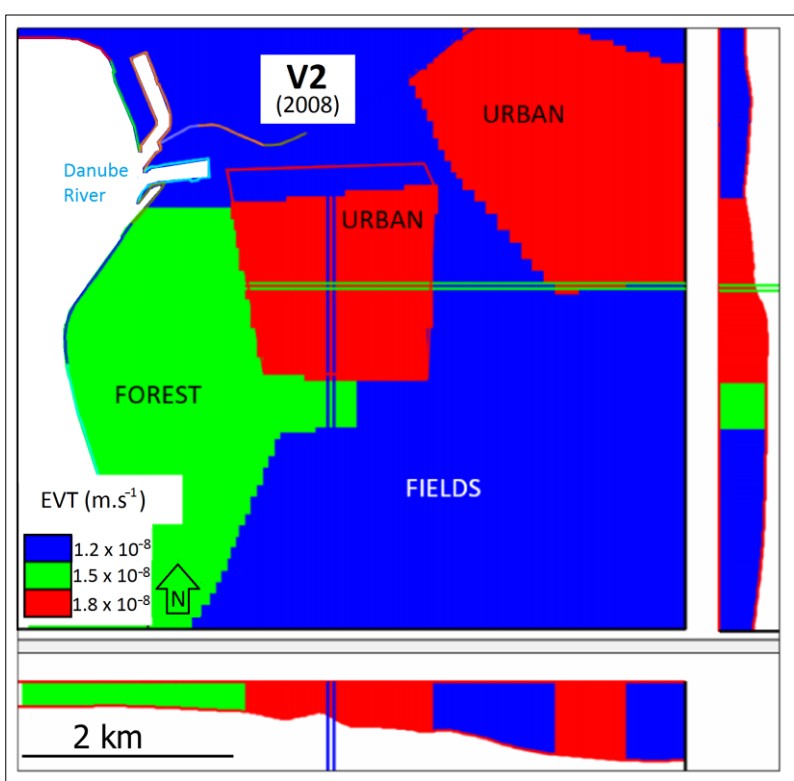

**Figure 7.** Evapotranspiration zonation with the optimized values within V2 model variant (planar view and vertical cross sections along the blue and green lines).

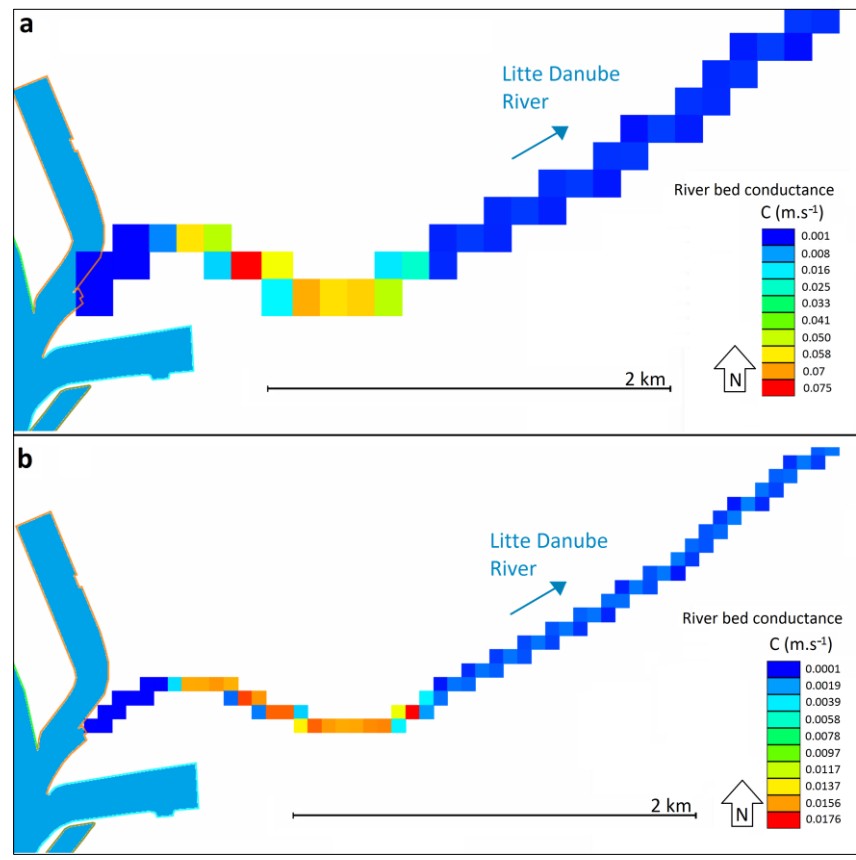

**Figure 8.** Optimized Little Danube riverbed conductance (C) within V3 (**a**) and V4 (**b**) model variants.

## 3. Results

The resulting groundwater table for the V4 variant is depicted in Figure 9. The groundwater table within all model variants is approximately the same in the main characteristics (GW gradients and flow directions). Differences, however, are in the accurateness of the simulated groundwater level compared to observations at a relatively small scale.

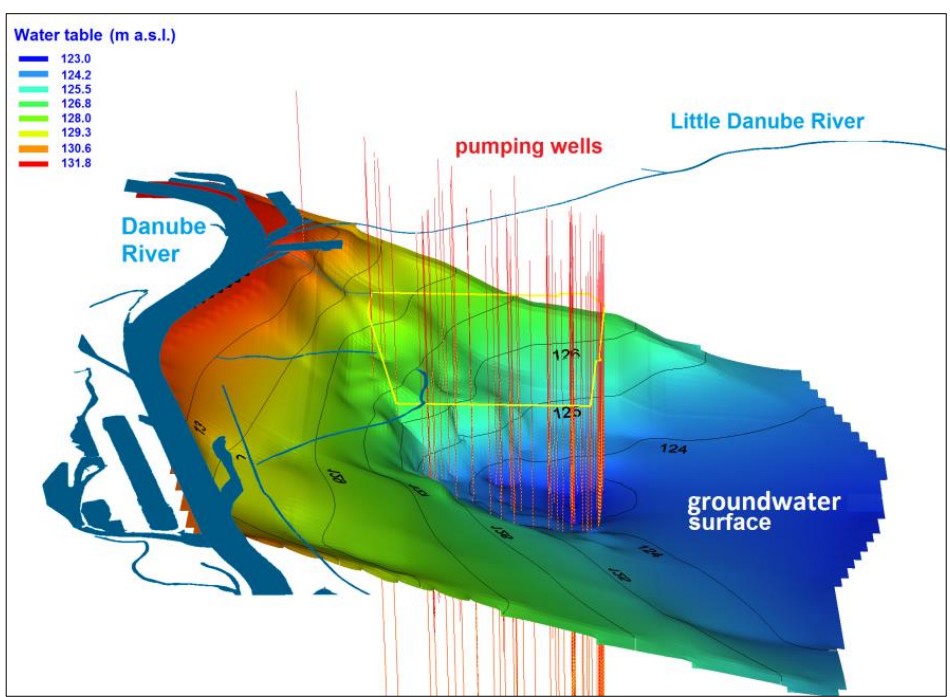

**Figure 9.** Resulting groundwater table of the V4 model variant, hydroizohypses, bodies of surface water, refinery area border, and pumping wells' location in the 3D modeled site view.

For **V1 calibration**, the resulting K values are shown in Figure 6a. They range from 0.005 m·s$^{-1}$ to $1 \times 10^{-8}$ m·s$^{-1}$. The variant V1 can be characterized as the worst one among all variants in each of the evaluated criteria listed in Table 2. In the scatter plot of OBS vs. SIM (Figure 10), the V1 variant performed relatively poorly within the calibration. The spatial distribution of residuals is not random (Figure 11) and the ratio of RMSE and OBS dispersion is 4.0% (Figure 12). The lowest value of AIC, AICc, and BIC criteria (Table 2) is achieved due to the extremely low number of calibrated parameters. Regarding the mentioned information criteria, the V1 scenario is assumed to provide the best prediction accuracy.

**Table 2.** Calibration characteristics of individual model variants.

| Model Variant | nPAR | OF | AVG RES (m) | AVG ABS RES (m) | RMSE (m) | AIC | AICc | BIC | RMSE and OBS Dispersion Ratio (%) |
|---|---|---|---|---|---|---|---|---|---|
| V1 | 2 | 64.5 | 0.04 | 0.25 | 0.34 | 69 | 69 | 77 | 4.0 |
| V2 | 43 | 26.5 | −0.05 | 0.14 | 0.22 | 113 | 121 | 298 | 2.6 |
| V3 | 139 | 11.0 | 0.05 | 0.11 | 0.14 | 289 | 389 | 885 | 1.6 |
| V4 | 255 | 5.8 | −0.01 | 0.07 | 0.1 | 516 | 977 | 1611 | 1.2 |

Notes: nPAR—number of parameters in model; OF—objective function; AVG RES—average of groundwater head residuals; AVG ABS RES—AVG of RES absolute values; RMSE—root mean square error; AIC, AICc, BIC—information criteria used; OBS—observed heads.

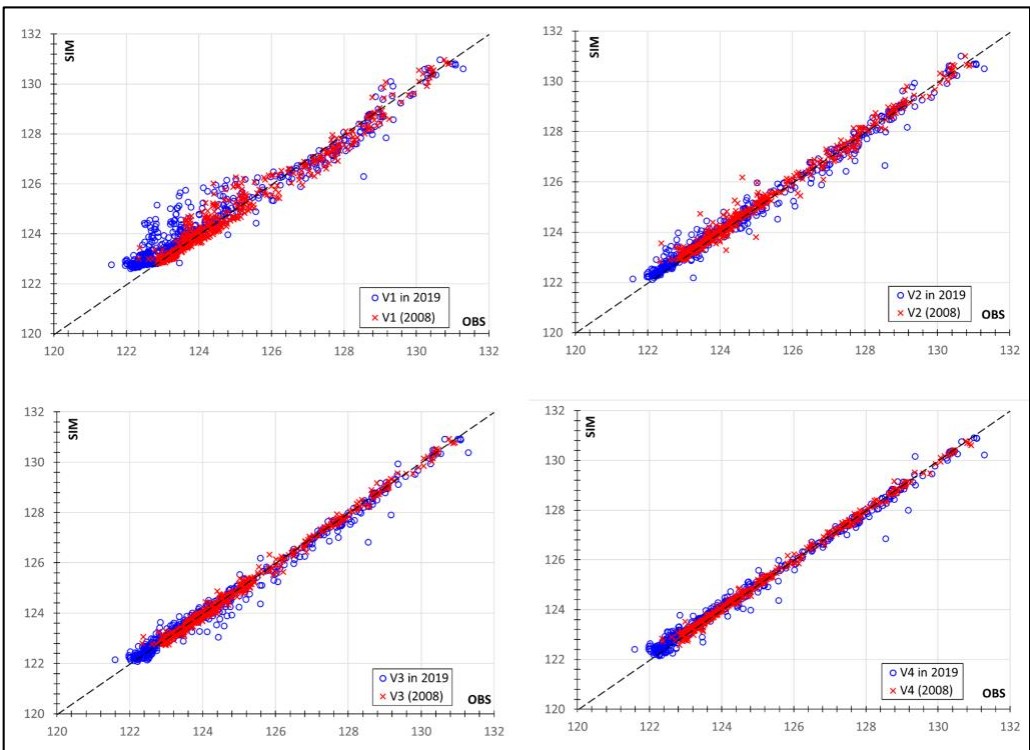

**Figure 10.** Scatterplots of observed and simulated groundwater heads within individual model variants (V1–V4) after calibration (2008) and within prediction (2019).

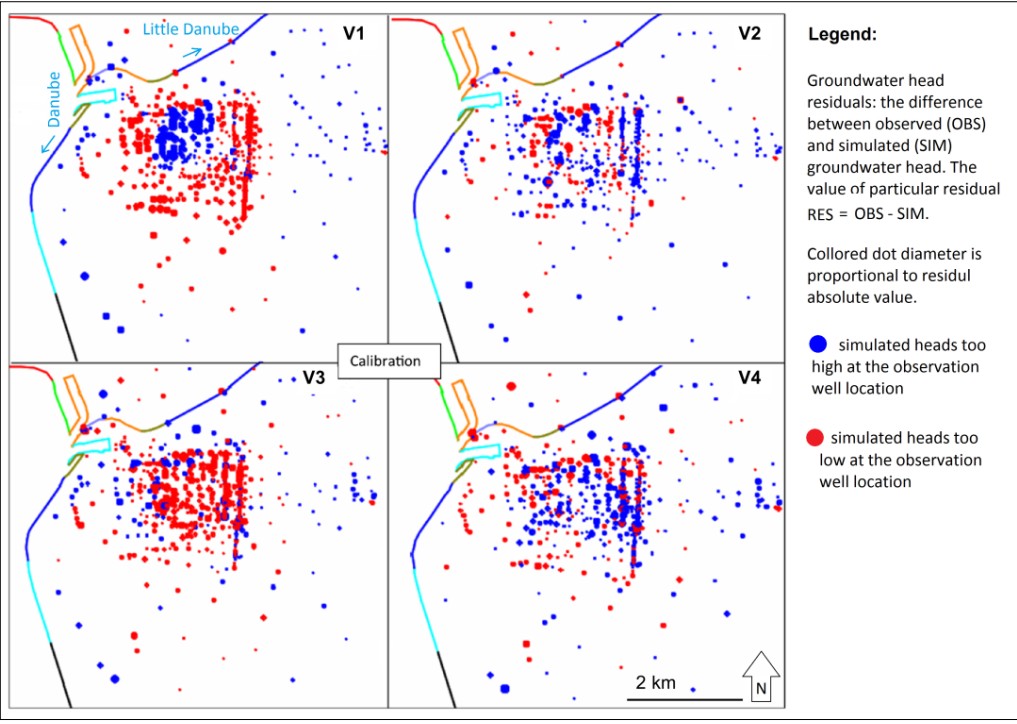

**Figure 11.** Groundwater head residuals within individual model variants (V1–V4) after calibration; blue: simulated heads too high, red: simulated heads too low.

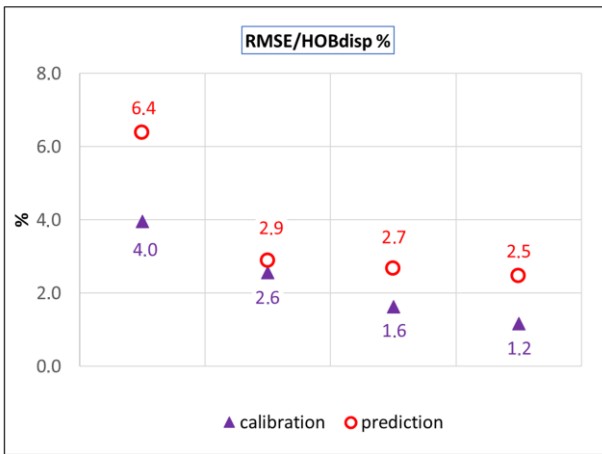

**Figure 12.** RMSE of groundwater head residuals and observed head dispersion (HOBdisp%) ratio of model variants (V1–V4) for calibration and prediction.

In **V1 prediction** performance, the value of all evaluated characteristics, which are introduced in Table 3, are the worst from all evaluated variants. In the scatterplot of OBS vs. SIM (Figure 10), the V1 variant performed relatively poorly within calibration for 2008, and even worse in the prediction for 2019. Residual distribution maps (Figure 13) show significant grouping of negative and positive residuals. The residual spatial distribution cannot be classified as random. The ratio of RMSE and OBS dispersion is 6.4% (Table 3, Figure 12). In all evaluated characteristics, the V1 model performed significantly worse in prediction compared to its calibration fit. Despite the best values of AIC, AICc, and BIC information criteria, the V1 prediction performance is the worst among evaluated scenarios. Considering the results, the V1 variant can be considered an example of conceptual oversimplification.

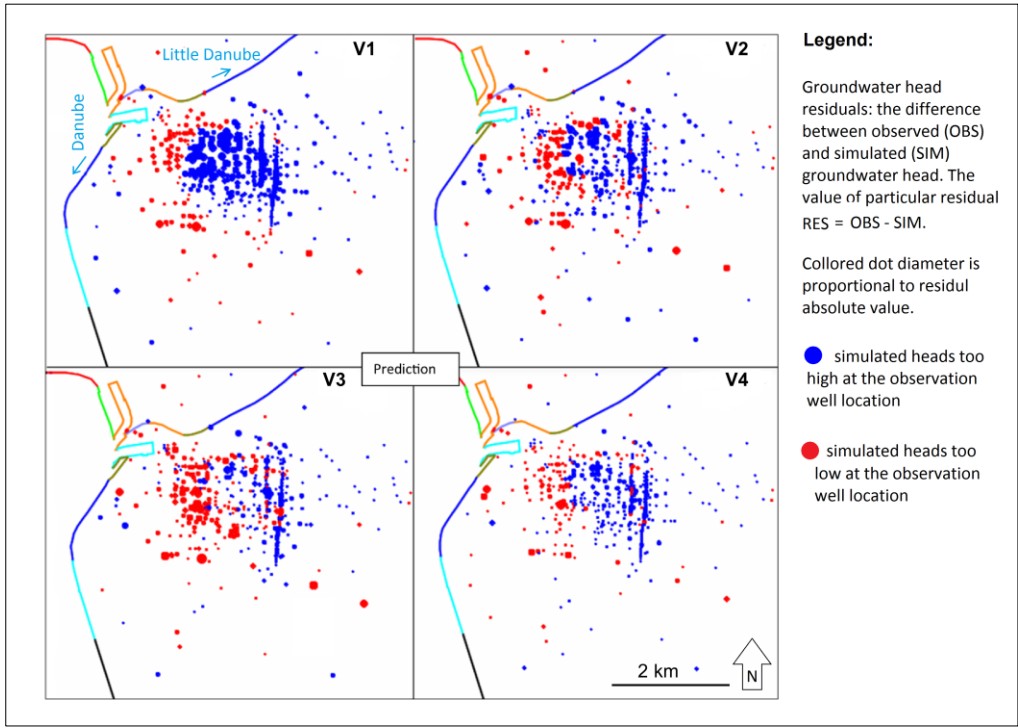

**Figure 13.** Groundwater head residuals within individual model variants (V1–V4) for prediction; blue: simulated heads too high, red: simulated heads too low.

**Table 3.** Prediction characteristics of individual model variants.

| Model Variant | OF | AVG RES (m) | AVG ABS RES (m) | RMSE (m) | RMSE and OBS Dispersion Ratio (%) |
|---|---|---|---|---|---|
| **V1_2019** | 205.7 | −0.29 | 0.45 | 0.62 | 6.4 |
| **V2_2019** | 44.2 | −0.05 | 0.20 | 0.28 | 2.9 |
| **V3_2019** | 35.7 | 0.03 | 0.17 | 0.26 | 2.7 |
| **V4_2019** | 31.4 | −0.06 | 0.17 | 0.24 | 2.5 |

Notes: OF—objective function; AVG RES—average of groundwater head residuals; AVG ABS RES—AVG of RES absolute values; RMSE—root mean square error; OBS—observed heads.

For **V2 calibration**, the resulting distribution of K values is shown in Figure 6b. They range from 0.05 m·s$^{-1}$ to $1 \times 10^{-8}$ m·s$^{-1}$. The resulting EVT values are shown in Figure 7. EVT in urbanized areas reached the highest value of 567 mm·y$^{-1}$. Here, it is likely that a significant effect of the interception and drainage of precipitation from the artificial surfaces of the area takes place. In the forests, the EVT reached a calibrated value of 473 mm·y$^{-1}$. In agriculture fields, the EVT had the lowest value of 378 mm·y$^{-1}$. OF and RMSE values are significantly lower compared to V1 (Table 2). This result represents a significantly better overall fit of the higher parametrized scenario V2 over V1 within the calibration. In the scatterplot of OBS vs. SIM (Figure 10), the V2 variant performs significantly better than V1. The spatial distribution of residuals is partly random and partly grouped (Figure 11). The ratio of RMSE and OBS dispersion is 2.6% (Figure 12). The relatively favorable values of AIC, AICc, and BIC criteria (Table 2) are achieved due to the relatively low number of calibrated parameters at a relatively low value of OF. From the V1 and V2 comparison, where the conceptual difference lies in the zonal calibration of the Quaternary aquifer K and zonal calibration of EVT, it can be concluded that the effect of higher parametrization has a significant impact on the overall model fit. Regarding the AIC, AICc, and BIC evaluation results, V2 is the second most successful calibration scenario.

**V2 prediction** performance is significantly better than the V1 model (Table 3), but still worse than its calibration fit. This is evident from the scatterplot of OBS vs. SIM (Figure 10). Residual distribution maps (Figure 13) show significant grouping of negative and positive residuals. The residual spatial distribution is not random and is worse in the prediction than in the calibration. The ratio of RMSE and OBS dispersion is 2.9%, which is slightly higher than for the calibration (Figure 12). In all the evaluated characteristics, the V2 model performs worse in prediction compared to its calibration fit.

For **V3 calibration**, the resulting distribution of K values is shown in Figure 6c. They range from 0.05 m·s$^{-1}$ to $1 \times 10^{-11}$ m·s$^{-1}$. The overall improvement in the calibration fit against the V1 and V2 models was indicated by the statistics introduced in Table 2. In the scatterplot of OBS vs. SIM (Figure 10), the V3 variant performs slightly better than the V2 variant. The spatial distribution of residuals is still not random (Figure 11) and is even worse than in the case of the V2 scenario. The V3 solution represents a ratio of RMSE and OBS dispersion of 1.6% (Table 2, Figure 12). This is the best value among the evaluated variants so far. The calibrated riverbed conductance of the Little Danube River is shown in Figure 8a. It ranges between the values 0.075 m·s$^{-1}$ and $2.12 \times 10^{-12}$ m·s$^{-1}$. From the AIC, AICc, and BIC evaluation, it follows that the V3 calibration variant is significantly less accurate for prediction than the V2 and V1 scenarios, due to the increased number of parameters and less significant OF value reduction (Table 2). Despite a reduction in the computation grid density, the value of RMSE, OF, and other related criteria are better than in the previous scenarios. The input precipitation and evapotranspiration difference of $4.76 \times 10^{-9}$ m·s$^{-1}$ (150 mm·y$^{-1}$), which was applied as one parameter for the whole model domain (the initial value is regional, not exact for local areas), has been optimized to $4.6 \times 10^{-9}$ m·s$^{-1}$ (145 mm·y$^{-1}$).

Within the **V3 prediction** performance, all evaluated statistics are slightly better than in the case of the V2 scenario; however, from a practical point of view, the prediction

accuracies of V2 and V3 are very similar. The prediction performance of V3 is worse than its calibration fit. In the scatter plot of OBS vs. SIM (Figure 10), V3 performs similar to the V2 variant. Residual distribution maps (Figure 13) show a significant grouping of negative and positive residuals, as well as in the V2 case. The ratio of RMSE and OBS dispersion is 2.7% (Figure 12).

For **V4 calibration**, the resulting distribution of K values is shown in Figure 6d. They range from 0.05 m·s$^{-1}$ to $1 \times 10^{-11}$ m·s$^{-1}$. The increased parametrization leads to increased calibration accuracy, which is the best among the evaluated scenarios. The improvement was recorded in all the evaluated characteristics (Table 2), and the RMSE reaches 0.1 m. In the scatterplot of OBS vs. SIM (Figure 10), V4 provides the best solution. Spatial distribution of residuals is close to random (Figure 11). The ratio of RMSE and OBS dispersion is 1.2% (Table 2, Figure 12). The resulting riverbed conductance of the Little Danube is shown in Figure 8b. From the AIC, AICc, and BIC evaluation point of view, the V4 variant is the least probable (Table 2) due to a significant increase in the number of parameters and a relatively slight decrease in OF and RMSE.

**V4 prediction** performance is the best, but is similar to V2 and V3 (Table 3). In the scatterplots of OBS vs. SIM (Figure 10), the V4 scenario prediction performs the best, as well as in the calibration stage. Residual distribution randomness (Figure 13) is worse than in V2 and quite similar to V3. The V4 model performs significantly worse in prediction compared to the calibration stage.

Particularly interesting is the greater improvement of model prediction accuracy between the V1 and V2 models compared to the improvement in model calibration fit between the V1 and V4 models (Figure 12). The finer K zonation (V3 and V4 models) overcompensated the absence of EVT zonation (applied only in the V2 model) within the impact on the calibration and prediction accuracy. Flow path analysis (Figure 14) shows that similarly successful scenarios (by prediction accuracy) can generate different GW pathlines. This is especially the case in the V3 model, in which the relatively low grid density has an impact on the shape of the predicted groundwater pathlines. The V1 model produces the most conservative (if GW is polluted) pathlines, e.g. the virtual particles propagate furthest between the pumping wells in both (stop or pass-through) representations of the "weak sinks" using MODPATH 6 code.

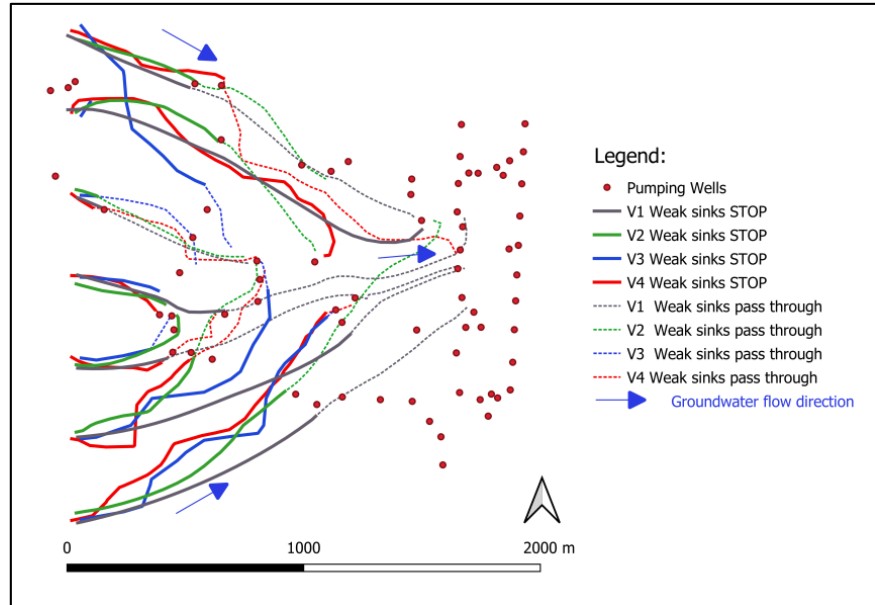

**Figure 14.** Groundwater pathlines prediction of individual model variants with the stop and pass-through weak sinks setting in the MODPATH 6 software.

In the calibration process, the difference between particular information criteria values becomes larger with an increasing number of parameters. The most penalizing criterion is the BIC followed by AICc and AIC. At the same time, the overall model fit with observation improves. Within prediction performance, a near constant accuracy of the V2, V3, and V4 models was recorded. This can be an indication of unproper K zonation, which does not represent the real K distribution with a near-constant structural error effect.

## 4. Discussion

Larger numbers of zones may lead to a better calibration fit with observation, but too many zones may lead to poor parameter estimation [32]. This also corresponds to the claim of [33] that no matter which regularization methodology is employed (e.g. zonation), the inevitable consequence of its use is a loss of detail in the calibrated field. This, in turn, can lead to erroneous predictions made by a model that is "well-calibrated". Additionally, the "unknown unknowns" in addition to the "known unknowns" always exist, which deteriorate the prediction accuracy [34]. Ref. [35], seeking different model conceptualizations within the Generalized Likelihood Uncertainty Estimation—Bayesian Model Averaging methods (GLUE—MBA), considered any spatial distribution (zonation) of a hydraulic conductivity field obtained through proper calibration as a valid representation of the K field. Since modelers usually know almost nothing about the exact distribution of K, this approach can then be assumed relevant. Ref. [36] claimed that with the increase in extrapolation, conceptual uncertainty also increases. The author of [37] in [38] demonstrated that the values estimated for lumped parameters can only be interpreted as the outcomes of a user-specified averaging process of pertinent system properties. The lumping of parameters (e.g., replacement of a continuous property field by a small number of zones of piecewise properties) introduces a structural error [38]. From the K-zonation point of view, the more K zones, the closer the resemblance to a continuum and a better calibration and prediction accuracy can be expected. In the current study, the overall calibration fit and real prediction accuracy are consistent with this conclusion.

The imperfection in the presented models may also be due to the temporal structure (measurement frequency) of the observation data and boundary condition data with average values for the calibration period, as well as the various frequencies of GW head measurement for various parts of the model area and lack of detailed information in the RCH and EVT distribution. Moreover, the spatial distribution of head observation points is far from uniform.

In [13], where the horizontal K was zonally calibrated, the information criterion BIC selected the simplest scenario as the best. This result also corresponds with our results. Ref. [14] found that the values of the AICc, BIC, and GCV statistics suggest that only the homogeneous model is clearly inferior, revealing that variations in K are important. Ref. [15] performed zonal calibration of K in which the most complex scenario had the best fit to observation data in the calibration stage. The AIC, AICc, and BIC criteria selected the simplest model as the most probable. These results also correspond with our findings.

The simplest scenario in the presented work can be considered (based on the post-audit of prediction accuracy results) as the oversimplification example. This statement is similar to the conclusion of [20] that neither very simple nor very complex models are likely to provide the most accurate predictions. However, no increased level of complexity that leads to worse prediction accuracy was found in the presented work. Starting from the V2 variant stage, the build-up of more complex models (V3, V4) resulted in a better fit in calibration but, at the same time, to almost zero improvement in prediction accuracy; thus, from a practical point of view, it can be seen as a meaningless effort.

The principal contribution of the presented study is the given picture of the parametrization influence on calibration fit and on the real prediction accuracy of the models at the study site. Since the hydrogeological measurements have been continually performed at this site for decades, the presented procedure of model prediction accuracy assessment can be repeated for various calibration and prediction time periods to verify or widen the pre-

sented results and to obtain more general views. The broader or more specifically focused investigation findings can lead to savings in parametrization efforts within new models and to better future optimization. Especially interesting would be the comparison of the resulting catchment areas of the pumping wells or the whole GWHP system between individual model scenarios and the field measurements. These analyses are crucial for predicting the impact of possible changes (for example, new pumping wells or different pumping rates at existing wells) in GWHP system operation with the resulting system optimization.

## 5. Conclusions

In this study performed at the Slovnaft site close to Bratislava in southwestern Slovakia contaminated by petroleum products, the calibration fit and prediction accuracy of four model variants with gradually increasing parametrization from the V1 to V4 models were evaluated in the post-audit procedure. The principal factor considered in parametrization was the complexity of K zonation.

In terms of objective function in the calibration process, the best value of 5.8 was recorded in the most parametrized V4 model scenario. Within prediction, the best objective function was reached in the same scenario with the value of 31.4. The objective function improved with increasing parametrization in calibration as well as in the prediction stage. The same development can be seen in all numerical and graphical evaluations performed.

In contrast, the information criteria AIC, AICc, and BIC increased significantly with model parameterization. The "penalization" for the number of parameters here was much more significant than the improvement of the calibration fit. From the performed study, it follows that the prediction accuracy in terms of the calibration fit should be the highest in the V4 model. In terms of the information criteria, the prediction accuracy should be highest in the case of the simplest V1 scenario. The real prediction accuracy assessment revealed that the information criteria provided an inaccurate evaluation.

The residuals' RMSE to observed head dispersion ratio within the calibration stage reached 4.0%, 2.6%, 1.6%, and 1.2% in the V1, V2, V3, and V4 models, respectively. The same indicator in the prediction stage reached 2.9%, 2.7%, and 2.5% in the V2, V3, and V4 models, respectively. The simplest V1 scenario with the value of 6.4% was significantly worse. The slight prediction improvement from the V2 to the V4 scenario reveals ineffective parametrization in the V3 and the V4 scenarios from the prediction accuracy perspective.

The procedure of manual refinement of K-field zonation based on the groundwater level residuals' distribution reveals a high efficiency for obtaining a better fit within the model calibration. However, within prediction performance, this procedure was not effective after an intermediate level of parametrization (V2 scenario). The prediction accuracy remained almost the same in more parametrized scenarios (V3 and V4 models). From the perspective of all related circumstances summarized as calibration effort vs. accuracy of prediction, the V2 scenario with a medium level of parametrization can be considered as the best solution of the study.

Flow path analysis showed that similarly successful scenarios (based on their prediction accuracy) can generate different groundwater pathlines. This is especially so in the case of the V3 model, in which the lower grid density has an impact on the shape of predicted groundwater pathlines. The simplest model scenario produces the most conservative outcome with respect to pathline propagation and possible spreading of groundwater contamination.

Based on the study results, there can be the following recommendations:

- the K-field zonation based on groundwater level residuals' distribution can be valuable in the calibration process if there are only limited K data from the field survey;
- higher parametrization does not necessarily lead to a more effective solution regarding prediction accuracy and several variants of a solution with continual post-audit evaluation should be used whenever possible;
- different model variants with similar prediction accuracy in terms of groundwater level fit can produce different groundwater pathlines; and, finally,

- the information criteria AIC, AICc, and BIC can be inaccurate in the evaluation of model prediction accuracy.

**Author Contributions:** M.Z.: Investigation, Conceptualization, Modeling, Writing—original draft, D.K.: Supervision, Formal analysis, Funding acquisition, K.H.: Review and editing, O.S.: Review and editing, Formal analysis, Š.M.: Investigation, Resources, Validation, Ľ.D.: Investigation, Data curation, A.B.: Investigation, Data curation. All authors have read and agreed to the published version of the manuscript.

**Funding:** We would like to thank the Slovak Research and Development Agency, which financially supported the research under contract No. APVV–14–0174 as well as the Ministry of Education, Science, Research and Sport of the Slovak Republic under contract No. VEGA 1/0302/21.

**Data Availability Statement:** The data presented in this study are available on request from the corresponding author. The data are not publicly available due to privacy.

**Acknowledgments:** We would like to thank the representatives of the SLOVNAFT, a.s. and the VÚRUP, a.s. companies for providing all data and enabling this research to be carried out.

**Conflicts of Interest:** The authors declare no conflict of interest. The funders had no role in the design of the study; in the collection, analyses, or interpretation of data; in the writing of the manuscript; or in the decision to publish the results.

## Abbreviations

| | |
|---|---|
| AIC | Akaike Information Criterion |
| AICc | Corrected Akaike Information Criterion |
| AVG | average |
| AVG ABS RES | averaged absolute residuals |
| BIC | Bayesian Information Criterion |
| E, W, N, S | east, west, north, south |
| EVT | evapotranspiration or evapotranspiration package/module in MODFLOW-2005 program |
| f | function |
| GLUE-MBA | Generalized Likelihood Uncertainty Estimation–Bayesian Model Averaging methods |
| GW | groundwater |
| GWHP | Groundwater Hydraulic Protection System |
| H | hydraulic head |
| CHD | Time-Variant Specified-Head package/module in MODFLOW-2005 program |
| K | hydraulic conductivity ($m \cdot s^{-1}$) |
| Kx, Ky | horizontal hydraulic conductivity ($m \cdot s^{-1}$) in "x" and "y" direction, respectively |
| Kz | vertical hydraulic conductivity ($m \cdot s^{-1}$) |
| LPF | layer property flow package/module in MODFLOW-2005 program |
| nPAR | number of adjusted parameters during calibration |
| OBS | observed groundwater head |
| OF | sum of squared residuals or objective function |
| PCG | preconditioned conjugate gradient package (solver) in MODFLOW-2005 program |
| Q | discharge or pumping rate ($m^3 . s^{-1}$) |
| Q pump | overall pumping rate at modeled site |
| RES | groundwater head residual (difference between observed and simulated head) |
| RCH | recharge or recharge package/module in MODFLOW-2005 program |
| RIV | river package/module in MODFLOW-2005 program |
| RMSE | root mean square error |
| SIM | calculated (simulated) groundwater head |
| V1–V4 | model scenarios (variants) |
| WEL | well package/module in MODFLOW-2005 program |

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
