# Peer review of "The Impact of Groundwater Model Parametrization on Calibration Fit and Prediction Accuracy—Assessment in the Form of a Post-Audit at the SLOVNAFT Oil Refinery Site, in Slovakia"

_water, doi:10.3390/w15050839_

Round 1

Reviewer 1 Report

Contribution is interesting, to my opinion too many abbreviations.

Corrections:

line 10 ... should be Faculty of Science ,...

line 36 ... the figure over the Keywords???

line 37 ... keywords separated by dots? (semicolon or comma)

line 104 ... 39 km2 ?

line 132 ... 5.10-2 ?

line 134 ... Ky = Kxy/5 ... why? (no reference)

line 148 ... figure 3 ... should be Quaternary

line 150 ... should be better (vertical exaggeration 1:20)

line 246 ... figure 5 ... explain the differenece in red and blue dots.

line 267 ... should be Horizontal.

line 263 ... The EVT value depends on groundwater level depth under the surface. There is no mention about that data (everywher is just the altitude!). Do you think it has any sence to consider it (value on the line 290) for groundwater flow modeling? 

line 377 ... 1.10-11 ...?

Explain, why did you choose the years 2008 (calibration) and 2019 (prediction). Are they somehow specific from hydrological point of view?

Round 2

Reviewer 2 Report

The introduction and literature needs to improvements and more study about the study problem

Author Response

We thank reviewer for the comments which helped to improve our manuscript.

The introduction and literature has been improved.